# ROBUST ATTRIBUTIONS REQUIRE RETHINKING ROBUSTNESS METRICS

## ABSTRACT

For machine learning models to be reliable and trustworthy, their decisions must be interpretable. As these models find increasing use in safety-critical applications, it is important that not just the model predictions but also their explanations (as feature attributions) be robust to small human-imperceptible input perturbations. Recent works have shown that many attribution methods are fragile and have proposed improvements in either the attribution methods or the model training. Existing works measure attributional robustness by metrics such as top-$k$ intersection, Spearman's rank-order correlation (or Spearman's $\rho$) or Kendall's rank-order correlation (or Kendall's $\tau$) to quantify the change in feature attributions under input perturbation. However, we show that these metrics are fragile. That is, under such metrics, a simple random perturbation attack can seem to be as significant as more principled attributional attacks. We instead propose Locality-sENSitive (LENS) improvements of the above metrics, namely, LENS-top-$k$, LENS-Spearman and LENS-Kendall, that incorporate the locality of attributions along with their rank order. Our locality-sensitive metrics provide tighter bounds on attributional robustness and do not disproportionately penalize attribution methods for reasonable local changes. We show that the robust attribution methods proposed in recent works also reflect this premise of locality, thus highlighting the need for a locality-sensitive metric for progress in the field. Our empirical results on well-known benchmark datasets using well-known models and attribution methods support our observations and conclusions in this work.

## 1 INTRODUCTION

There has been an explosive increase in the use of deep neural network (DNN)-based models for many applications in recent years, which has resulted in an equivalently increasing interest in finding ways to interpret the decisions made by these models. Interpretability is an important aspect of responsible and trustworthy AI, and attribution methods are important for explaining and debugging real-world AI/ML systems. Attribution methods are used across application domains today (see (Gade et al., 2020) for a general discussion and (Tang et al.; Yap et al.; Oviedo et al., 2022; Oh & Jeong, 2020) for some examples), despite their limitations. These methods (Zeiler et al., 2010; Simonyan et al., 2014; Bach et al., 2015; Selvaraju et al., 2017; Chattopadhyay et al., 2018; Sundararajan et al., 2017; Shrikumar et al., 2016; Smilkov et al., 2017; Lundberg & Lee, 2017) find approaches to explain the decisions made by these models, instead of using them as a black box. With growing numbers of explanation methods (see (Lipton, 2018; Samek et al., 2019; Fan et al., 2021; Zhang et al., 2020; Zhang & Zhu, 2018) for surveys), there have also been recent concerted efforts on analyzing and proposing methods to ensure the robustness of DNN model explanations. This requires that the model explanations (also known as attributions) do not change with human-imperceptible changes in input (Chen et al., 2019; Sarkar et al., 2021). For example, an explanation code for a credit card failure cannot change significantly for a small human-imperceptible change in input features, or the saliency maps explaining the risk prediction of a chest X-ray should not change significantly with a minor human-imperceptible change in the image. This is referred to as attributional robustness.

From another perspective, DNN-based models are known to have a vulnerability to imperceptible adversarial perturbations (Biggio et al., 2013; Szegedy et al., 2014), which make them misclassify input images. These small imperceptible perturbations are constructed using techniques like Fast Gradient Signed Method (FGSM) (Goodfellow et al., 2015) and Projected Gradient Descent (PGD) (Madry et al., 2018). Adversarial training with PGD is a well-known solution to obtain

better adversarial robustness to attacks like FGSM and PGD. While adversarial robustness has received significant attention over the last few years (Ozdag, 2018; Silva & Najafirad, 2020), attributional robustness has received lesser attention. In an early effort, (Ghorbani et al., 2019) provided a method to construct a small imperceptible perturbation which when added to the input $x$ will lead to commonly used correlation measures such as top-$k$ intersection, Spearman's rank-order correlation or Kendall's rank-order correlation used to quantify the change between the explanation map of the original image and that of the perturbed image to drop (see Figure 1).

Defenses against such attributional attacks have been proposed recently in (Chen et al., 2019; Singh et al., 2020; Wang et al., 2020; Sarkar et al., 2021), which focus on regularizing the loss function (Chen et al., 2019; Singh et al., 2020; Sarkar et al., 2021) or use smoothing techniques on gradients (Wang et al., 2020) to reduce the impact of the attack.

Across all the efforts so far (Ghorbani et al., 2019; Chen et al., 2019; Singh et al., 2020; Wang et al., 2020; Sarkar et al., 2021), the robustness of attributions on input perturbation is measured using metrics such as top-$k$ intersection, and rank correlations like Spearman's $\rho$ and Kendall' $\tau$ to estimate the quality of the attack. While such metrics give a reasonable estimate when there are significant changes in attributions (see Figure 1 *row 1*), they are highly sensitive to minor local changes in attributions, even by one or few pixel coordinate locations (see Figure 1 *row 2*). We, in fact, show (Section 3.1) that under such metrics, a random perturbation is as strong an attack as existing benchmark methods such as (Ghorbani et al., 2019). This may not be a true indicator of the robustness of attributions of a model, and thereby misleading to research efforts that build on current observations. Beyond highlighting this important issue, we instead propose locality-sensitive improvements of the above metrics that incorporate the locality of attributions along with their rank order. We show that such a locality-sensitive distance is upper-bounded by a metric based on symmetric set difference. Our key contributions are summarized below:

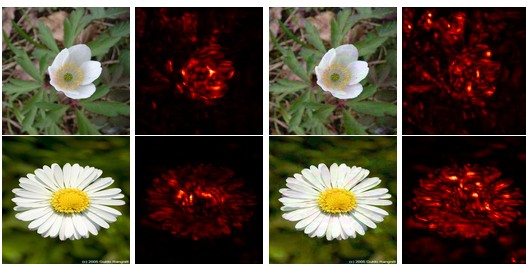

Figure 1: Attributional attack on Flower dataset using Ghorbani et al. (2019) method on a ResNet model. Columns 1 and 3 show the image before and after an imperceptible perturbation; Columns 2 and 4 show the corresponding attributions. Note the change in attributions despite no perceptible change in input. **Row 1** shows a distinct change in top-$k$ pixels with the highest attribution; **Row 2** shows only a *local change* in top-$k$ pixels with the highest attribution still within the object. The intersection between the top-1000 pixels before and after perturbation is less than 0.16 in both the cases; thus, as a metric it cannot really distinguish the two.

- We firstly observe that existing robustness metrics for model attributions overpenalize minor drifts in attribution, leading to a false sense of fragility. We go on to show that under existing such metrics, a random perturbation is as good an attack as principled methods like (Ghorbani et al., 2019).

- In order to address this issue, we propose Locality-sENSitive (LENS) improvements of existing metrics, namely, LENS-top-$k$, LENS-Spearman and LENS-Kendall, that incorporate the locality of attributions along with their rank order. Our locality-sensitive metrics do not disproportionately penalize attribution methods for reasonable local changes.

- We show that our proposed LENS variants are well-motivated by metrics defined on the space of attributions, and they provide tighter bounds on the attributional robustness of already known improvements in attribution methods and model training designed for better attributions.

- Our comprehensive empirical results on benchmark datasets and models used in existing work clearly support our aforementioned observations, and support the need for the LENS variants of the metrics.

- We also show that existing robust attribution methods implicitly reflect this premise of locality, thus highlighting the need for a locality-sensitive metric for progress in the field.

## 2 BACKGROUND AND RELATED WORK

We herein discuss background literature related to our work from three different perspectives: a brief summary of explanation/attribution methods in general, review of recent work in attributional robustness (both attacks and defenses), and other recent related work.

**Attribution Methods.** Existing efforts on explainability in DNN models can be broadly categorized as: local and global methods, model-agnostic and model-specific methods, or as post-hoc and ante-hoc (intrinsically interpretable) methods (Molnar, 2019; Lecue et al., 2021). Almost all the popular methods in use today – including methods to visualize weights and neurons (Simonyan et al., 2014; Zeiler & Fergus, 2014), guided backpropagation (Springenberg et al., 2015), CAM (Zhou et al.), GradCAM (Selvaraju et al., 2017), Grad-CAM++ (Chattopadhyay et al., 2018), LIME (Ribeiro et al., 2016), DeepLIFT (Shrikumar et al., 2016; 2017), LRP (Bach et al., 2015), Integrated Gradients (Sundararajan et al., 2017), SmoothGrad (Smilkov et al., 2017)), DeepSHAP (Lundberg & Lee, 2017) and TCAV (Kim et al., 2018) – are all post-hoc methods, which are used on top of a pre-trained DNN model as a separate layer/module to explain its predictions. We focus on such post-hoc attribution methods in this work. For a more detailed survey of explainability methods for DNN models, please see (Lecue et al., 2021; Molnar, 2019; Samek et al., 2019).

**Robustness of Attributions.** With the growing numbers of attribution methods, there has also been a recent focus on identifying the desirable characteristics of such methods (Alvarez-Melis & Jaakkola, 2018; Adebayo et al., 2018; Yeh et al., 2019; Chalasani et al., 2020; Tomsett et al., 2020; Boggust et al., 2022; Agarwal et al., 2022). A key desired trait that has been highlighted by many of these efforts is robustness or stability of attributions, i.e., the explanation should not vary significantly within a small local neighborhood of the input (Alvarez-Melis & Jaakkola, 2018; Chalasani et al., 2020). Ghorbani et al. (2019) showed that well-known methods such as gradient-based attributions, DeepLIFT (Shrikumar et al., 2017) and Integrated Gradients (IG) (Sundararajan et al., 2017) are vulnerable, and provided an algorithm to construct a small imperceptible perturbation which when added to the input results in change in the attribution. Slack et al. (2020) showed that methods like LIME (Ribeiro et al., 2016) and DeepSHAP (Lundberg & Lee, 2017) too are vulnerable to such manipulations. The identification of such vulnerability and attack has subsequently led to multiple research efforts that have attempted to make a model's attributions robust. Chen et al. (2019) proposed a regularization-based approach, where an explicit regularizer term is added to the loss function to maintain the model gradient across input (IG, in particular) while training the DNN model. This was subsequently extended by (Sarkar et al., 2021; Singh et al., 2020; Wang et al., 2020), all of whom provide different training strategies and regularizers to improve attributional robustness of models. Each of these methods including (Ghorbani et al., 2019) measures change in attribution before and after input perturbation using top-$k$ intersection, and/or rank correlations like Spearman's $\rho$ and Kendall' $\tau$. Such metrics have recently, in fact, further been used to understand issues surrounding attributional robustness (Wang & Kong, 2022). Other efforts that quantify stability of attributions in tabular data also use Euclidean distance or its variants (Alvarez-Melis & Jaakkola, 2018; Yeh et al., 2019; Agarwal et al., 2022). Each of these metrics look for dimension-wise correlation or pixel-level matching between attribution maps before and after perturbation, and thus penalize even a minor change in attribution (say, even by one pixel coordinate location). This may not be reasonable, and could even be misleading. In this work, we highlight the need to revisit such metrics, and propose a locality-sensitive variant that can be easily integrated into all existing metrics.

**Other Related Work.** In other related efforts that have studied similar properties of attribution-based explanations, Carvalho et al. (2019); Bhatt et al. (2020) stated that stable explanations should not vary too much between similar input samples, unless the model's prediction changes drastically. All the abovementioned attributional attacks and defenses (Ghorbani et al., 2019; Sarkar et al., 2021; Singh et al., 2020; Wang et al., 2020) maintain this property, since they focus on input perturbations that change the attribution without changing the model prediction itself. Similarly, Arun et al. (2020) and Fel et al. (2022) introduced the notions of repeatability/reproducibility and generalizability respectively, both of which focus on the desired property that a trustworthy explanation must point to similar evidence across similar input images. In this work, we provide a practical metric to study this notion of similarity by considering locality-sensitive metrics.

## 3 ATTRIBUTIONAL ROBUSTNESS METRICS ARE FRAGILE

We first discuss the fragility of existing metrics, before presenting our locality-sensitive variants. The robustness of an attribution method has generally been measured in earlier work (Ghorbani et al., 2019; Chen et al., 2019) by computing the similarity between attributions of an original input and the attribution of the same input with perturbations, and averaging this similarity across multiple input images. Similar to adversarial perturbations (Madry et al., 2018), such "attributional perturbations" are carefully constructed attack vectors of small $\ell_\infty$ norm that do not change the model prediction but only the attributions on the perturbed inputs (Ghorbani et al., 2019). Using this, Ghorbani et al.

(2019) show that many popular attribution methods are fragile. The common similarity measures between attributions are top-$k$ intersection and rank correlation coefficients such as Spearman's $\rho$ and Kendall's $\tau$. Note that all of the above similarity measures depend only on the rank order of features in the attributions (e.g., rank order of pixels in images).

## 3.1 RANDOM VECTORS ARE ATTRIBUTIONAL ATTACKS UNDER EXISTING METRICS

Random vectors of a small $\ell_\infty$ norm are often used as baselines of input perturbations (both in adversarial robustness (Silva & Najafirad, 2020) and attributional robustness literature Ghorbani et al. (2019)), since it is known that predictions of neural network models are known to be resilient to random perturbations of inputs. Previous work by Ghorbani et al. (2019) has shown random perturbations to be a reasonable baseline to compare against their attributional attack. Extending it further, we show that a single input-agnostic random perturbation happens to be an effective *universal* attributional attack if we measure attributional robustness using a weak metric based on top-$k$ intersection. In other words, considering even a random perturbation happens to be a good attributional attack under such metrics, we show that existing metrics for attributional robustness such as top-$k$ intersection are extremely fragile, i.e., they would unfairly deem many attribution methods as fragile.

Integrated Gradients (IG) is a well-known attribution method based on well-defined axiomatic foundations (Sundararajan et al., 2017), which is commonly used in attributional robustness literature (Chen et al., 2019; Sarkar et al., 2021). We take a naturally trained CNN model on MNIST and perturb the images using a random perturbation (an independent random perturbation per input image) as well as a single, input-agnostic or *universal* random perturbation for all images. Figure 2 shows a sample image from the MNIST dataset and the visual difference between the IG of the original image, the IG after adding a random perturbation, and the IG after adding a *universal* random perturbation. The IG after the universal random attack (Figure 2d) is visually more dissimilar to the IG of the original image (Figure 2b) than the IG of a simple random perturbation (Figure 2c). (Note that top-$k$ intersection between Figure 2b and 2c is only 0.62, although the two look similar. As stated in the caption, a locality-sensitive metric shows them to be closer in attribution however.)

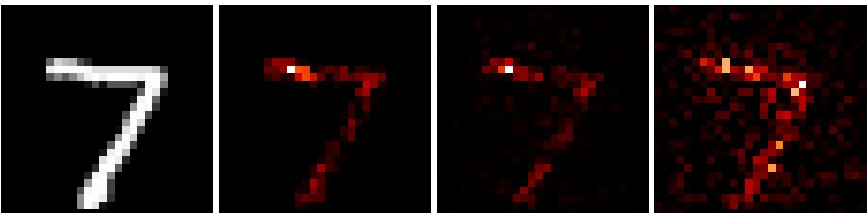

(a) Original image    (b) IG(original image)    (c) IG after random    (d) IG after universal

Figure 2: Sample image from MNIST shows that the Integrated Gradients (IG) after a universal random perturbation are more dissimilar than IG after a simple, independent random perturbation for each input. All perturbations have random $\pm 1$ coordinates, scaled down to have $\ell_\infty$ norm $\epsilon = 0.3$. (c) has a top-$k$ intersection of 0.68, while (d) has a top-$k$ intersection of 0.62. With our locality-sensitive metric, (c) has LENS-top-$k$ of 0.99 and (d) has LENS-top-$k$ of 1.0.

Similarly, Table 1 shows that under existing metrics to quantify attributional robustness of IG on a naturally trained CNN model, even a single, input-agnostic or *universal* random perturbation can sometimes be a more effective attributional attack than using an independent random perturbation for each input. The detailed description of our experimental setup can be found in Appendix B.

## 4 LOCALITY-SENSITIVE METRICS FOR ATTRIBUTIONAL ROBUSTNESS

Section 3 raises the need to look beyond current metrics used to study attributional robustness. Current analyses of attributional attacks use similarity measures such as top-$k$ intersection and rank correlation metrics such as Spearman's $\rho$, Kendall's $\tau$ that ignore where the top attributions are located in the image. As an aside, these are not *metrics* in the mathematical sense but theoretically interesting metrics have been derived from them in the ranking literature (Fagin et al., 2003).

## 4.1 DEFINING LOCALITY-SENSITIVE METRICS FOR ATTRIBUTIONS

We propose a natural way to extend the existing similarity measures to further incorporate the locality of pixel attributions in images to derive more robust, locality-sensitive measures of attributional

Table 1: Attributional robustness of IG on naturally trained models measured using average top-$k$ intersection, Spearman's $\rho$ and Kendall's $\tau$ between IG(original image) and IG(perturbed image). $k = 100$ for MNIST, Fashion MNIST, GTSRB and $k = 1000$ for Flower. **Bold** entries indicate where the universal random perturbation is a stronger attributional attack.

| Dataset | Perturbation | top-$k$ intersection | Spearman's $\rho$ | Kendall's $\tau$ |
|---|---|---|---|---|
| MNIST | random | 0.7500 | 0.5347 | 0.4337 |
| | universal random | 0.5855 | 0.4831 | 0.4063 |
| Fashion MNIST | random | 0.5385 | 0.6791 | 0.5152 |
| | universal random | 0.5280 | 0.7154 | 0.5688 |
| GTSRB | random | 0.8216 | 0.9433 | 0.8136 |
| | universal random | 0.9293 | 0.9887 | 0.9243 |
| Flower | random | 0.8202 | 0.9562 | 0.8340 |
| | universal random | 0.9344 | 0.9908 | 0.9321 |

robustness. Let $a_{ij}(x)$ denote the attribution value or importance assigned to the $(i, j)$-th pixel in an input image $x$, and let $S_k(x)$ denote the set of $k$ pixel positions with the largest attribution values. Let $N_w(i, j) = \{(p, q) : i - w \leq p \leq i + w, \ j - w \leq q \leq j + w\}$ be the neighboring pixel positions within a $(2w + 1) \times (2w + 1)$ window around the $(i, j)$-th pixel. By a slight abuse of notation, we use $N_w(S_k(x))$ to denote $\bigcup_{(i,j) \in S_k(x)} N_w(i, j)$, that is, the set of all pixel positions that lie in the union of $(2w + 1) \times (2w + 1)$ windows around the top-$k$ pixels.

For a given attributional perturbation $\text{Att}(\cdot)$, let $T_k = S_k(x + \text{Att}(x))$ denote the top-$k$ pixels in attribution values after applying the attributional perturbation $\text{Att}(x)$. Top-$k$ intersection metric outputs $|S_k(x) \cap T_k(x)| / k$. We propose *Locality-sENSitive top-$k$ metrics* (LENS-top-$k$) as $|N_w(S_k(x)) \cap T_k(x)| / k$ and $|S_k(x) \cap N_w(T_k(x))| / k$, along the lines of precision and recall used to evaluate ranking. We define Locality-sENSitive Spearman's $\rho$ (LENS-Spearman) and Locality-sENSitive Kendall's $\tau$ (LENS-Kendall) metrics as rank correlation coefficients for the smoothed ranking orders according to $\tilde{a}_{ij}(x)$'s and $\tilde{a}_{ij}(x + \text{Att}(x))$'s, respectively. These can be used to compare two different attributions for the same image, the same attribution method on two different images, or even two different attributions on two different images, as long as the attribution vectors lie in the same space, e.g., images of the same dimensions where attributions assign importance values to pixels.

We show some theoretically interesting properties of locality-sensitive measures below. Let $\mathbf{a}_1$ and $\mathbf{a}_2$ be two attribution vectors for two images, and let $S_k$ and $T_k$ be the set of top $k$ pixels in these images according to $\mathbf{a}_1$ and $\mathbf{a}_2$, respectively. We define a locality-sensitive top-$k$ distance between two attribution vectors $\mathbf{a}_1$ and $\mathbf{a}_2$ as $d_k^{(w)}(\mathbf{a}_1, \mathbf{a}_2) \stackrel{\text{def}}{=} \text{prec}_k^{(w)}(\mathbf{a}_1, \mathbf{a}_2) + \text{recall}_k^{(w)}(\mathbf{a}_1, \mathbf{a}_2)$, where

$$\text{prec}_k^{(w)}(\mathbf{a}_1, \mathbf{a}_2) \stackrel{\text{def}}{=} \frac{|S_k \setminus N_w(T_k)|}{k} \quad \text{and} \quad \text{recall}_k^{(w)}(\mathbf{a}_1, \mathbf{a}_2) \stackrel{\text{def}}{=} \frac{|T_k \setminus N_w(S_k)|}{k},$$

similar to precison and recall used in ranking literature, except the key difference being the inclusion of neighborhood items based on locality. We state below a monotonicity property of $d_k^{(w)}(\mathbf{a}_1, \mathbf{a}_2)$ and upper bound it in terms of the symmetric set difference of top-$k$ attributions.

**Proposition 1.** *For any $w_1 \leq w_2$, we have $d_k^{(w_2)}(\mathbf{a}_1, \mathbf{a}_2) \leq d_k^{(w_1)}(\mathbf{a}_1, \mathbf{a}_2) \leq |S_k \triangle T_k| / k$, where $\triangle$ denotes the symmetric set difference, i.e., $A \triangle B = (A \setminus B) \cup (B \setminus A)$.*

Combining $d_k^{(w)}(\mathbf{a}_1, \mathbf{a}_2)$ across different values of $k$ and $w$, we can define a distance

$$d(\mathbf{a}_1, \mathbf{a}_2) = \sum_{k=1}^{\infty} \alpha_k \sum_{w=0}^{\infty} \beta_w \, d_k^{(w)}(\mathbf{a}_1, \mathbf{a}_2),$$

where $\alpha_k$ and $\beta_w$ be non-negative weights, monotonically decreasing in $k$ and $w$, respectively, such that $\sum_k \alpha_k < \infty$ and $\sum_w \beta_w < \infty$. We show that the distance defined above is upper bounded by a metric that is similar to the metrics proposed in Fagin et al. (2003) based on symmetric set difference of top-$k$ ranks to compare two rankings.

**Proposition 2.** *$d(\mathbf{a}_1, \mathbf{a}_2)$ defined above is upper bounded by $u(\mathbf{a}_1, \mathbf{a}_2)$ given by*

$$u(\mathbf{a}_1, \mathbf{a}_2) = \sum_{k=1}^{\infty} \alpha_k \sum_{w=0}^{\infty} \beta_w \, \frac{|S_k \triangle T_k|}{k},$$

*and $u(\mathbf{a}_1, \mathbf{a}_2)$ defines a bounded metric on the space of attribution vectors.*

Note that top-$k$ intersection, Spearman's $\rho$ and Kendall's $\tau$ do not take the attribution values $a_{ij}(x)$'s into account but only the rank order of pixels according to these values. We also define a locality-sensitive $w$-smoothed attribution as follows.

$$\tilde{a}_{ij}^{(w)}(x) = \frac{1}{(2w+1)^2} \sum_{\substack{(p,q) \in N_w(i,j), \\ 1 \leq p,q \leq n}} a_{pq}(x)$$

We show that the $w$-smoothed attribution leads to a contraction in the $\ell_2$ norm commonly used in theoretical analysis of simple gradients as attributions.

**Proposition 3.** *For any inputs $x, y$ and any $w \geq 0$, $\left\| \tilde{\mathbf{a}}^{(w)}(x) - \tilde{\mathbf{a}}^{(w)}(y) \right\|_2 \leq \|\mathbf{a}(x) - \mathbf{a}(y)\|_2$.*

Thus, any theoretical bounds on the attributional robustness of simple gradients in $\ell_2$ norm proved in previous works continue to hold for locality-sensitive $w$-smoothed gradients. For example, Wang et al. (2020) show the following Hessian-based bound on simple gradients. For an input $x$ and a classifier or model defined by $f$, let $\nabla_x(f)$ and $\nabla_y(f)$ be the simple gradients w.r.t. the inputs at $x$ and $y$. Theorem 3 in Wang et al. (2020) upper bounds the $\ell_2$ distance between the simple gradients of nearby points $\|x - y\|_2 \leq \delta$ as $\|\nabla_x(f) - \nabla_y(f)\|_2 \lesssim \delta \lambda_{\max}(H_x(f))$, where $H_x(f)$ is the Hessian of $f$ w.r.t. the input at $x$ and $\lambda_{\max}(H_x(f))$ is its maximum eigenvalue. By Proposition 3 above, the same continues to hold for $w$-smoothed gradients, i.e., $\left\| \tilde{\nabla}_x^{(w)}(f) - \tilde{\nabla}_y^{(w)}(f) \right\|_2 \lesssim \delta \lambda_{\max}(H_x(f))$. The proofs of all the propositions above are included in Appendix A.

### 4.2 LOCALITY-SENSITIVE (LENS) ATTRIBUTIONAL ROBUSTNESS METRICS ARE STRONGER

The top-$k$ intersection is a measure of similarity instead of distance. Therefore, in our experiments for attributional robustness, we use locality-sensitive similarity measures $w$-LENS-prec@$k$ and $w$-LENS-recall@$k$ to denote $1 - \text{prec}_k^{(w)}(\mathbf{a}_1, \mathbf{a}_2)$ and $1 - \text{recall}_k^{(w)}(\mathbf{a}_1, \mathbf{a}_2)$, respectively, where $\mathbf{a}_1$ is the attribution of the original image and $\mathbf{a}_2$ is the attribution of the perturbed image. For rank correlation coefficients such as Kendall's $\tau$ and Spearman's $\rho$, we compute $w$-LENS-Kendall and $w$-LENS-Spearman as the same Kendall's $\tau$ and Spearman's $\rho$ but computed on the locality-sensitive $w$-smoothed attribution map $\tilde{\mathbf{a}}^{(w)}$ instead of the original attribution map $\mathbf{a}$. We also study how these similarity measures and their resulting attributional robustness measures change as we vary $k$ and $w$.

In this section, we measure the attributional robustness of Integrated Gradients (IG) on naturally trained models as top-$k$ intersection, $w$-LENS-prec@$k$ and $w$-LENS-recall@$k$ between the IG of the original images and the IG of their perturbations obtained by various attacks. The attacks we consider are the top-$t$ attack and the mass-center attack of Ghorbani et al. (2019) as well as random perturbation. All perturbations have $\ell_\infty$ norm bounded by $\delta = 0.3$ for MNIST, $\delta = 0.1$ for Fashion MNIST, and $\delta = 8/255$ for GTSRB and Flower datasets. The values of $t$ used to construct the top-$t$ attacks of Ghorbani et al. (2019) are $t = 200$ on MNIST, $t = 100$ on Fashion MNIST and GTSRB, and $t = 1000$ on Flower. In the robustness evaluations for a fixed $k$, we use $k = 100$ on MNIST, Fashion MNIST and GTSRB, and $k = 1000$ on Flower.

**Comparison of top-$k$ intersection, 1-LENS-prec@$k$ and 1-LENS-recall@$k$.** Figure 3 shows that top-$k$ intersection penalizes IG even for small, local changes. 1-LENS-prec@$k$ and 1-LENS-recall@$k$ values are always higher in comparison across all datasets in our experiments. Moreover, on both MNIST and Fashion MNIST, 1-LENS-prec@$k$ is roughly 2x higher (above 90%) compared to top-$k$ intersection (near 40%). In other words, an attack may appear stronger under a weaker measure of attributional robustness, if it ignores locality.

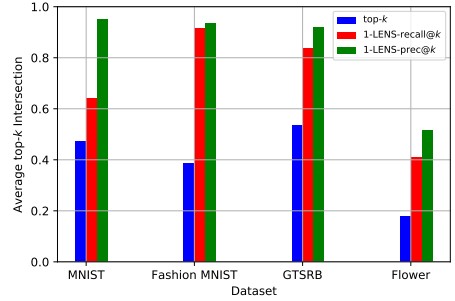

Figure 3: Attributional robustness of IG on naturally trained models measured as average top-$k$ intersection, 1-LENS-prec@$k$ and 1-LENS-recall@$k$ between IG(original image) and IG(perturbed image) obtained by the top-$t$ attack (Ghorbani et al., 2019) across different datasets.

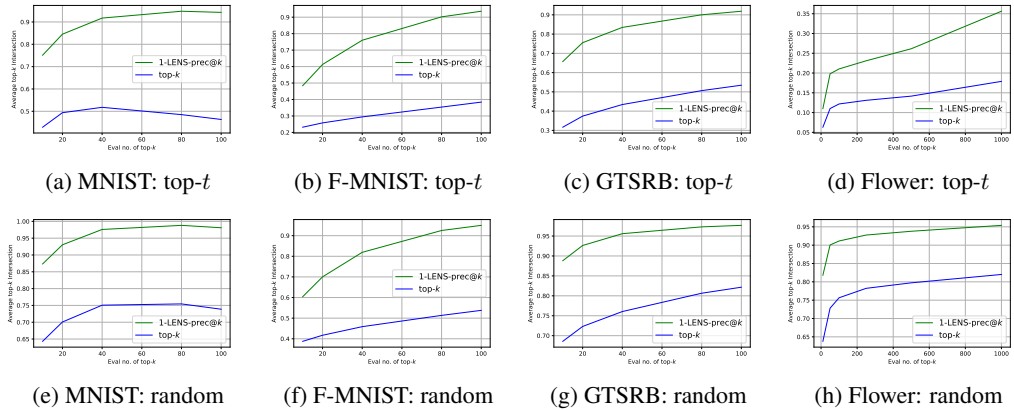

| (a) MNIST: top-$t$ | (b) F-MNIST: top-$t$ | (c) GTSRB: top-$t$ | (d) Flower: top-$t$ |
|---|---|---|---|
| (e) MNIST: random | (f) F-MNIST: random | (g) GTSRB: random | (h) Flower: random |

Figure 4: Attributional robustness of IG on naturally trained models measured as average top-$k$ intersection and 1-LENS-prec@$k$ between IG(original image) and IG(perturbed image). Perturbations are obtained by the top-$t$ attack (Ghorbani et al., 2019) and random perturbation. The plots show how the above measures change with varying $k$ across different datasets.

**The effect of varying** $k$. Figure 4 shows a large disparity between top-$k$ intersection and 1-LENS-prec@$k$ even when $k$ is large. Figure 4 shows that top-$k$ intersection can be very low even when the IG of the original and the IG of the perturbed images are locally very similar, as indicated by high 1-LENS-prec@$k$. Our observation holds for the perturbations obtained by the top-$t$ attack of (Ghorbani et al., 2019) as well as a random perturbation across all datasets in our experiments.

$w$-**LENS-prec@$k$ for varying** $w$. Figure 5 that $w$-LENS-prec@$k$ increases as we increase $w$ to consider larger neighborhoods around the pixels with top attribution values. This holds for multiple perturbations, namely, top-$t$ attack and mass-center attack by Ghorbani et al. (2019) as well as a random perturbation. Notice that the top-$t$ attack of Ghorbani et al. (2019) is constructed specifically for the top-$t$ intersection objective, and perhaps as a result, shows larger change when we increase local-sensitivity by increasing $w$ in the robustness measure.

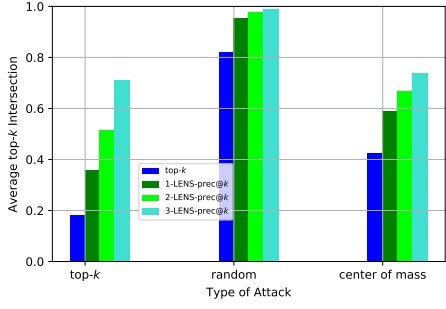

Figure 5: Attributional robustness of IG on naturally trained models measured as average top-$k$ intersection and $w$-LENS-prec@$k$ between IG(original image) and IG(perturbed image). Perturbations are obtained by the top-$t$ attack and the mass-center attack (Ghorbani et al., 2019) as well as random perturbation. The plots show the effect of varying $w$ on Flower dataset.

Figure 6: Attributional robustness of IG on naturally trained models measured as average Spearman's $\rho$, 1-LENS-Spearman, Kendall $\tau$ and 1-LENS-Kendall between IG(original image) and IG(perturbed image). The perturbations are obtained by the top-$t$ attack of Ghorbani et al. (2019).

**Comparison of Spearman's $\rho$ and Kendall's $\tau$ with 1-LENS-Spearman and 1-LENS-Kendall.**
Figure 6 compares Spearman's $\rho$ and Kendall's $\tau$ with 1-LENS-Spearman and 1-LENS-Kendall measures for attributional robustness. We observe that 1-smoothing of attribution maps increases the corresponding Kendall's $\tau$ and Spearman's $\rho$ measures of attributional robustness, and this observation holds across all datasets in our experiments. As a result, we believe that 1-LENS-Spearman and 1-LENS-Kendall result in better or tighter attributional robustnes measures than Spearman's $\rho$ and Kendall's $\tau$.

**Modifying the attack of Ghorbani et al. (2019) for** $1$**-LENS-prec@**$k$ **objective**    A natural question is whether the original top-$k$ attack of Ghorbani et al. (2019) seem weaker under locality-senstitive robustness measures only because the attack was specifically constructed for a corresponding top-$k$ intersection objective. Since the construction of the attack in Ghorbani et al. (2019) is modifiable for any similarity objective, we use $1$-LENS-prec@$k$ to construct a new attributional attack for $1$-LENS-prec@$k$ objective based on the $k \times k$ neighborhood of pixels. Surprisingly, we notice that it leads to a *worse* attributional attack, if we measure its effectiveness using the top-$k$ intersection; see Figure 7. In other words, attributional attacks against locality-sensitive measures of attributional robustness are non-trivial and may require fundamentally different ideas.

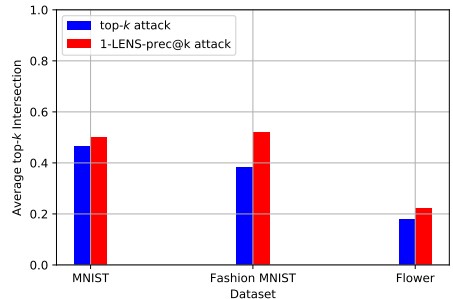

Figure 7: Average top-$k$ intersection between IG(original image) and IG(perturbed image) on naturally trained models where the perturbation is obtained by incorporating $1$-LENS-prec@$k$ objective in the Ghorbani et al. (2019) attack.

Appendix E contains additional results with similar conclusions when Simple Gradients are used instead of Integrated Gradients (IG) for obtaining the attributions.

## 5    CONNECTION TO ROBUST ATTRIBUTION TRAINING METHODS

A common approach to get robust attributions is to keep the attribution method unchanged but train the models differently in a way that the resulting attributions are more robust to small perturbations of inputs. Chen et al. (2019) proposed the first defense against the attributional attack of Ghorbani et al. (2019). Wang et al. (2020) also find that IG-NORM based training of Chen et al. (2019) gives models that exhibit attributional robustness against the top-$k$ attack of Ghorbani et al. (2019) along with adversarially trained models. Figure 8 shows a sample image from the Flower dataset, where the Integrated Gradients (IG) of the original image and its perturbation by the top-$k$ attack are visually similar for models that are either adversarially trained (trained using Projected Gradient De-

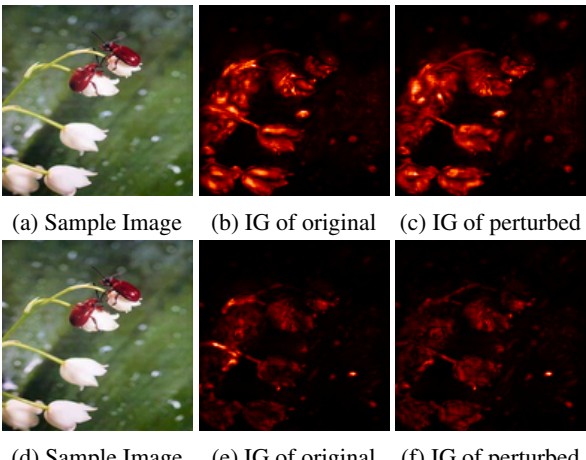

(a) Sample Image    (b) IG of original    (c) IG of perturbed

(d) Sample Image    (e) IG of original    (f) IG of perturbed

Figure 8: Sample image from Flower whose IG for PGD-trained (top) and IG-SUM-NORM trained (bottom) models seem robust to perturbations by the top-$k$ attack of Ghorbani et al. (2019).

scent or PGD-trained, as proposed by (Madry et al., 2018)) or IG-SUM-NORM trained as in Chen et al. (2019). In other words, these differently trained model guard the sample image against the attributional top-$k$ attack. Recent work by Nourelahi et al. (2022) has empirically studied the effectiveness of adversarially (PGD) trained models in obtaining better attributions, e.g., Figure 8b shows sharper attributions to features highlighting the ground-truth class.

Figure 9 shows that PGD-trained and IG-SUM-NORM trained models have more robust Integrated Gradients (IG) in comparison to their naturally trained counterparts, and this holds for the previously used measures of attributional robustness (e.g., top-$k$ intersection) as well as the new locality-sensitive measures we propose (e.g., $1$-LENS-prec@$k$, $1$-LENS-recall@$k$) across all datasets in our experiments. The top-$k$ attack of Ghorbani et al. (2019) is not a threat to IG if we simply measure its effectiveness using $1$-LENS-prec@$k$ (Figure 9(a-c) for MNIST, Fashion MNIST and GTSRB), and moreover, use IG on PGD-trained or IG-SUM-NORM trained models (Figure 9(d) for Flower).

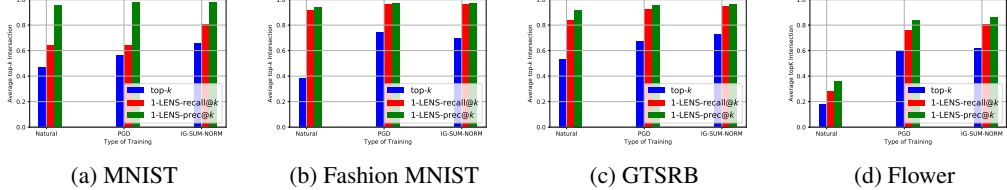

|  |  |  |  |
|---|---|---|---|
| (a) MNIST | (b) Fashion MNIST | (c) GTSRB | (d) Flower |

Figure 9: Average top-$k$ intersection, 1-LENS-prec@$k$, 1-LENS-recall@$k$ measured between IG(original image) and IG(perturbed image) for models that are naturally trained, PGD-trained and IG-SUM-NORM trained. The perturbation used is the top-$t$ attack of Ghorbani et al. (2019). Shown for (a) MNIST, (b) Fashion MNIST, (c) GTSRB and (d) Flower datasets.

The above observation about robustness of Integrated Gradients (IG) for PGD-trained and IG-SUM-NORM trained models holds even when we use 1-LENS-Spearman and 1-LENS-Kendall measures to quantify the attributional robustness to the top-$k$ attack of Ghorbani et al. (2019), and it holds across all datasets used in our experiments; see Figure 10. Moreover, the 1-LENS-Kendall and 1-LENS-Spearman values in Figure 10 are always higher than the corresponding Kendall's $\tau$ and Spearman's $\rho$ values, which further strengthen the conclusions from previous papers that IG on PGD-trained and IG-SUM-NORM trained models give better attributions.

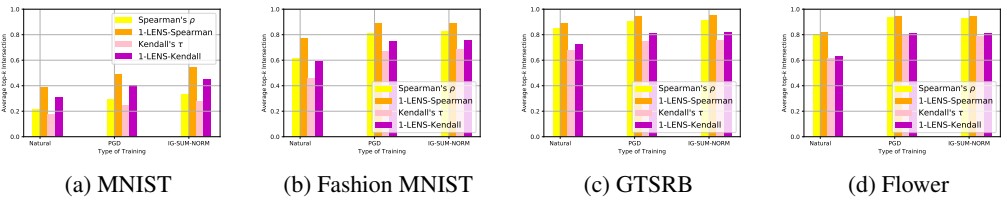

|  |  |  |  |
|---|---|---|---|
| (a) MNIST | (b) Fashion MNIST | (c) GTSRB | (d) Flower |

Figure 10: Average Kendall's $\tau$, Spearman's $\rho$, 1-LENS-Kendall and 1-LENS-Spearman used to measure the attributional robustness of IG on natrually trained, PGD-trained and IG-SUM-NORM trained models. The perturbation used is the top-$k$ attack of Ghorbani et al. (2019). Shown for (a) MNIST, (b) Fashion MNIST, (c) GTSRB and (d) Flower datasets.

Chalasani et al. (2020) show theoretically that $\ell_\infty$-adversarial training (PGD-training) leads to stable Integrated Gradients (IG) under $\ell_1$ norm. They also show empirically that PGD-training leads to sparse attributions (IG and DeepSHAP) when sparseness in measured indirectly as the change in the Gini index. Our empirical results extend their theoretical observation about stability of IG for PGD-trained models, as we measure local stability in terms of both the attribution values as well as their corresponding positions in the image.

## 6 CONCLUSION AND FUTURE WORK

We show that the fragility of attributions is an effect of using fragile robustness metrics such as top-$k$ intersection that only look at the rank order of attributions and fail to capture the closeness of pixel positions with high attributions. We highlight the need for locality-sensitive metrics for attributional robustness and propose some natural locality-sensitive extensions of the existing metrics. Theoretical understanding of locality-sensitive metrics of attributional robustness, constructing stronger attributional attacks for these metrics, and using them to build attributionally robust models are some important future directions.

**Reproducibility Statement.** The anonymous code for the paper can be found at this anonymous link.

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

SUPPLEMENTARY MATERIAL

The Appendix contains proofs, additional experiments to show that the trends hold across different datasets and other ablation studies which could not be included in the main paper due to space constraints.

## A    PROOFS FROM SECTION 4

We restate and prove Proposition 1 below.

**Proposition 4.** *For any $w_1 \leq w_2$, we have $d_k^{(w_2)}(\mathbf{a}_1, \mathbf{a}_2) \leq d_k^{(w_1)}(\mathbf{a}_1, \mathbf{a}_2) \leq |S_k \triangle T_k| / k$, where $\triangle$ denotes the symmetric set difference, i.e., $A \triangle B = (A \setminus B) \cup (B \setminus A)$.*

*Proof.* The inequalities follows immediately using $S \subseteq N_{w_1}(S) \subseteq N_{w_2}(S)$, for any $S$, and hence, $|S \setminus N_w(T)| \leq |S \setminus T|$, for any $S, T$ and $w$. □

We restate and prove Proposition 2 below.

**Proposition 5.** *$d(\mathbf{a}_1, \mathbf{a}_2)$ defined above is upper bounded by $u(\mathbf{a}_1, \mathbf{a}_2)$ given by*

$$u(\mathbf{a}_1, \mathbf{a}_2) = \sum_{k=1}^{\infty} \alpha_k \sum_{w=0}^{\infty} \beta_w \frac{|S_k \triangle T_k|}{k},$$

*and $u(\mathbf{a}_1, \mathbf{a}_2)$ defines a bounded metric on the space of attribution vectors.*

*Proof.* Proof follows from Proposition 1 and using the fact that symmetric set difference satisfies triangle inequality. □

We restate and prove Proposition 3 below.

**Proposition 6.** *For any inputs $x, y$ and any $w \geq 0$, $\left\| \tilde{\mathbf{a}}^{(w)}(x) - \tilde{\mathbf{a}}^{(w)}(y) \right\|_2 \leq \|\mathbf{a}(x) - \mathbf{a}(y)\|_2$.*

*Proof.*

$$\left\| \tilde{\mathbf{a}}^{(w)}(x) - \tilde{\mathbf{a}}^{(w)}(y) \right\|_2^2 = \sum_{1 \leq i,j \leq n} \left( \tilde{a}_{ij}^{(w)}(x) - \tilde{a}_{ij}^{(w)}(y) \right)^2$$

$$= \sum_{1 \leq i,j \leq n} \frac{1}{(2w+1)^4} \left( \sum_{\substack{(p,q) \in N_w(i,j), \\ 1 \leq p,q \leq n}} (a_{pq}(x) - a_{pq}(y)) \right)^2$$

$$\leq \sum_{1 \leq i,j \leq n} \frac{(2w+1)^2}{(2w+1)^4} \sum_{\substack{(p,q) \in N_w(i,j), \\ 1 \leq p,q \leq n}} (a_{pq}(x) - a_{pq}(y))^2 \text{ by Cauchy-Schwarz inequality}$$

$$= \frac{1}{(2w+1)^2} \sum_{1 \leq i,j \leq n} \sum_{\substack{(p,q) \in N_w(i,j), \\ 1 \leq p,q \leq n}} (a_{pq}(x) - a_{pq}(y))^2$$

$$\leq \frac{(2w+1)^2}{(2w+1)^2} \sum_{1 \leq p,q \leq n} (a_{pq}(x) - a_{pq}(y))^2$$

$$\quad \text{because each } (p,q) \text{ appears in at most } (2w+1)^2 \text{ possibles } N_w(i,j)\text{'s}$$

$$= \|\mathbf{a}(x) - \mathbf{a}(y)\|_2^2.$$

□

## B  DETAILS OF EXPERIMENTAL SETUP

The detailed description of the setup used in our experiments.

*Datasets:* We use the standard benchmark train-test split of all the datasets used in this work, that is publicly available. MNIST dataset consists of $70,000$ images of $28 \times 28$ size, divided into 10 classes: $60,000$ used for training and $10,000$ for testing. Fashion MNIST dataset consists of $70,000$ images of $28 \times 28$ size, divided into 10 classes: $60,000$ used for training and $10,000$ for testing. GTSRB dataset consists of $51,739$ images of $32 \times 32$ size, divided into 43 classes: $34,699$ used for training, $4,410$ for validation and $12,630$ for testing. Flower dataset consist of $1,360$ images of $128 \times 128$ size, divided into 17 classes: $1,224$ used for training and 136 for testing. GTSRB and Flower datasets were preprocessed exactly as given in Chen et al. (2019)[Appendix C] for consistency of results.

*Architectures:* For MNIST, Fashion MNIST, GTSRB and Flower datasets we use the exact architectures as used by Chen et al. (2019).

*Attribution robustness metrics:* We use the same comparison metrics as used by Ghorbani et al. (2019) and Chen et al. (2019) like top-$k$ pixels intersection, Spearman's $\rho$ and Kendall's $\tau$ rank correlation to compare attribution maps of the original and perturbed images. The $k$ value for top-$k$ attack along with settings like step size, number of steps and number of times attack is to be applied is as used by Chen et al. (2019) for the attack construction : MNIST(200,0.01,100,3), Fashion MNIST(100,0.01,100,3), GTSRB(100,1.0,50,3), Flower(1000,1.0,100,3).

*Sample sizes for attribution robustness evaluations:* *IG based experiments* For MNIST, Fashion MNIST and Flower with fixed top-$k$ attack similar to Chen et al. (2019) the complete test set were used to obtain the results. For GTSRB a random sample of size 1000 was used for all the experiments. *Simple gradient based experiments* For MNIST and Fashion MNIST a random sample of 2500/1000 from the test set. For GTSRB, a random sample of size 1000 and the complete test set for the Flower dataset.

*Adversarial training:* We use the standard setup as used by (Chen et al., 2019). We perform PGD based adversarial training with the provided $\epsilon$ budget using the following settings (number of steps, step size) for PGD : MNIST (40,0.01), Fashion MNIST(20,0.01), GTSRB(7,8/255), Flower(7,2/255).

*Training for Attributional Robustness:* We use the IG-SUM-NORM objective function for all the datasets study based on (Chen et al., 2019) based training. With the exact setting as given in paper with code [1].

*Hardware Configuration:* We used a server with 4 Nvidia GeForce GTX 1080i GPU and a server with 8 Nvidia Tesla V100 GPU to run the experiments in the paper.

---

[1]https://github.com/jfc43/robust-attribution-regularization

## C    THE FRAGILITY OF TOP-$k$ INTERSECTION

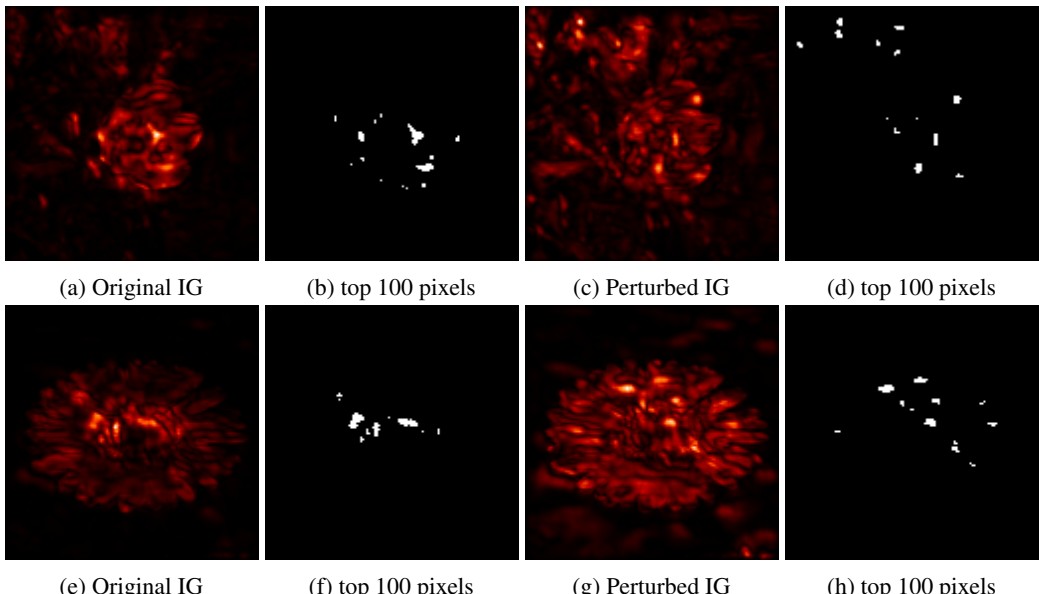

(a) Original IG    (b) top 100 pixels    (c) Perturbed IG    (d) top 100 pixels

(e) Original IG    (f) top 100 pixels    (g) Perturbed IG    (h) top 100 pixels

Figure 11: Sample top-$k$ highlighting using Flower.

# D EXPERIMENTS WITH INTEGRATED GRADIENTS

Below we present additional experimental results for Integrated Gradients (IG).

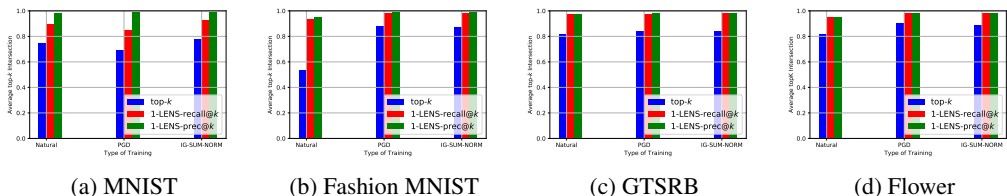

| (a) MNIST | (b) Fashion MNIST | (c) GTSRB | (d) Flower |

Figure 12: Attributional robustness of IG on naturally, PGD and IG-SUM-NORM trained models measured as top-$k$ intersection, 1-LENS-prec@$k$ and 1-LENS-recall@$k$ between the IG of the original images and the IG of their perturbations obtained by the random attack (Ghorbani et al., 2019) across different datasets.

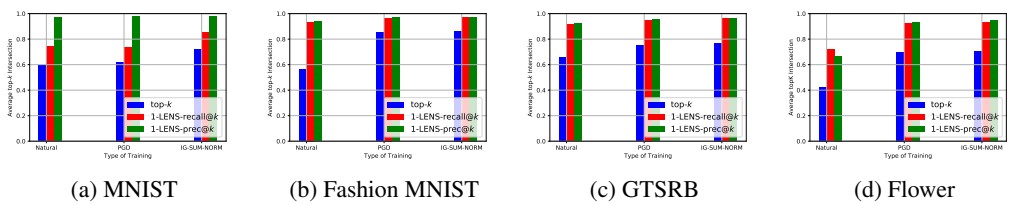

| (a) MNIST | (b) Fashion MNIST | (c) GTSRB | (d) Flower |

Figure 13: Attributional robustness of IG on naturally, PGD and IG-SUM-NORM trained models measured as top-$k$ intersection, 1-LENS-prec@$k$ and 1-LENS-recall@$k$ between the IG of the original images and the IG of their perturbations obtained by the mass-center attack (Ghorbani et al., 2019) across different datasets.

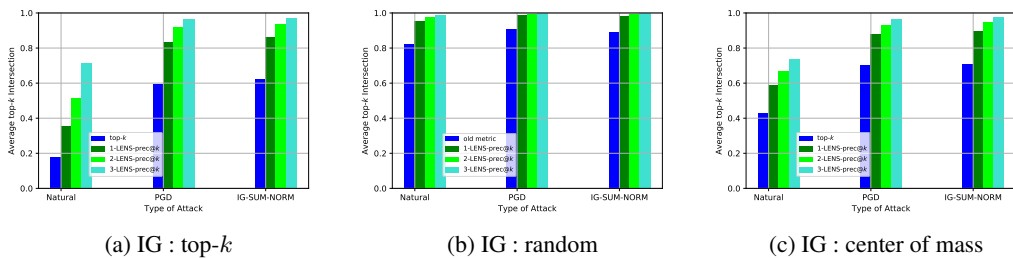

| (a) IG : top-$k$ | (b) IG : random | (c) IG : center of mass |

Figure 14: Attributional robustness of IG on naturally, PGD and IG-SUM-NORM trained models measured as top-$k$ intersection and $w$-LENS-prec@$k$ between the IG of the original images and the IG of their perturbations. Perturbations are obtained by the top-$t$ attack and the mass-center attack (Ghorbani et al., 2019) as well as a random perturbation. The plots show the effect of varying $w$ on Flower dataset.

# E EXPERIMENTS WITH SIMPLE GRADIENTS

We observe that our conclusions about Integrated Gradients (IG) continue to hold qualitatively, even if we replace IG with Simple Gradients as our attribution method.

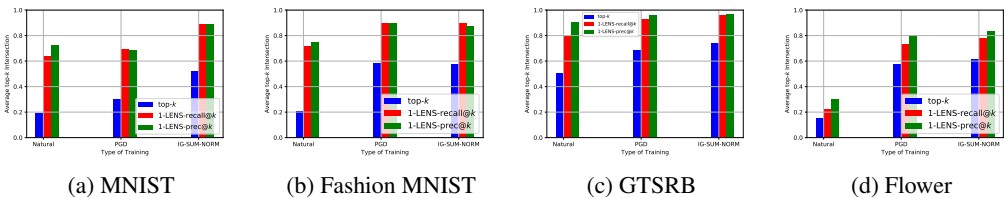

|   (a) MNIST   |   (b) Fashion MNIST   |   (c) GTSRB   |   (d) Flower   |
|---|---|---|---|

Figure 15: Attributional robustness of Simple Gradients on naturally, PGD and IG-SUM-NORM trained models measured as top-$k$ intersection, 1-LENS-prec@$k$ and 1-LENS-recall@$k$ between the Simple Gradient of the original images and the Simple Gradient of their perturbations obtained by the top-$k$ attack (Ghorbani et al., 2019) across different datasets.

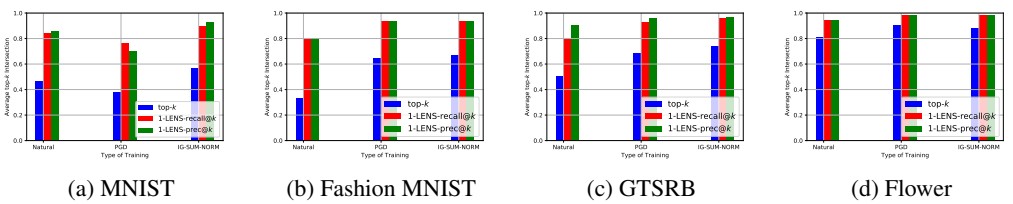

|   (a) MNIST   |   (b) Fashion MNIST   |   (c) GTSRB   |   (d) Flower   |
|---|---|---|---|

Figure 16: Attributional robustness of Simple Gradients on naturally, PGD and IG-SUM-NORM trained models measured as top-$k$ intersection, 1-LENS-prec@$k$ and 1-LENS-recall@$k$ between the Simple Gradient of the original images and the Simple Gradient of their perturbations obtained by the random attack (Ghorbani et al., 2019) across different datasets.

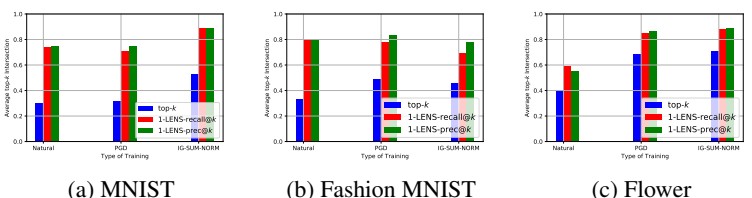

|   (a) MNIST   |   (b) Fashion MNIST   |   (c) Flower   |
|---|---|---|

Figure 17: Attributional robustness of Simple Gradients on naturally, PGD and IG-SUM-NORM trained models measured as top-$k$ intersection, 1-LENS-prec@$k$ and 1-LENS-recall@$k$ between the Simple Gradient of the original images and the Simple Gradient of their perturbations obtained by the mass-center attack (Ghorbani et al., 2019) across different datasets.

# F ADDITIONAL RESULTS FOR PGD-TRAINED AND IG-SUM-NORM TRAINED MODELS

Figure 21 and Figure 19 shows the impact of $k$ in top-$k$ for adversarially(PGD) trained and attributional(IG-SUM-NORM) trained network, respectively. But an important point to be noticed is that even with small number of features LENS is able to cross 70-80% which supports the observation of sparsity and stability of attributions achieved by adversarially(PGD) trained models by Chalasani et al. (2020). Similarly, the experiments with different $w$ value for $w$-LENS-top-$k$ in 20 clearly incidates that due to the stability properties at lower window sizes LENS is able to cross the 80% intersection quickly. Supporting that our metric nicely captures local stability very well. While we observed only the top-$k$ version of LENS. Figure 10 further extends the observation to LENS-Spearman and LENS-Kendall who to show that with LENS with a smoothing of $3 \times 3$ the

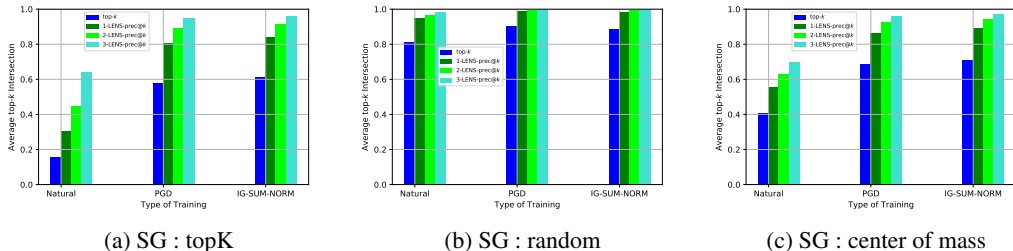

(a) SG : topK    (b) SG : random    (c) SG : center of mass

Figure 18: Attributional robustness of Simple Gradients on naturally, PGD and IG-SUM-NORM trained models measured as top-$k$ intersection and $w$-LENS-prec@$k$ between the IG of the original images and the IG of their perturbations. Perturbations are obtained by the top-$t$ attack and the mass-center attack (Ghorbani et al., 2019) as well as a random perturbation. The plots show the effect of varying $w$ on Flower dataset.

maps from adversarial and attributional robust models have a higher top-$k$ intersection above 70% in comparison to natural trained model.

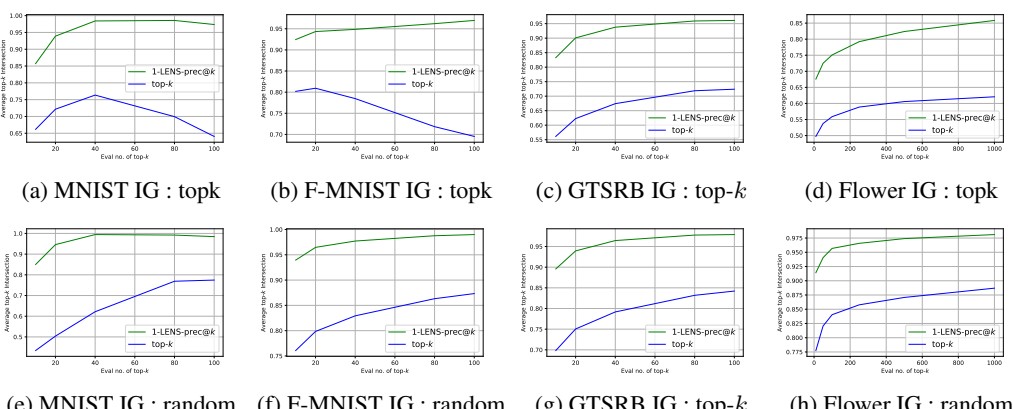

(a) MNIST IG : topk    (b) F-MNIST IG : topk    (c) GTSRB IG : top-$k$    (d) Flower IG : topk

(e) MNIST IG : random    (f) F-MNIST IG : random    (g) GTSRB IG : top-$k$    (h) Flower IG : random

Figure 19: Comparing the top-$k$ intersection between top-$k$ and 1-LENS-prec@k with different $k$ for top-$k$ evaluation while the attack is a fixed $k$. For IG-SUM-NORM trained network.

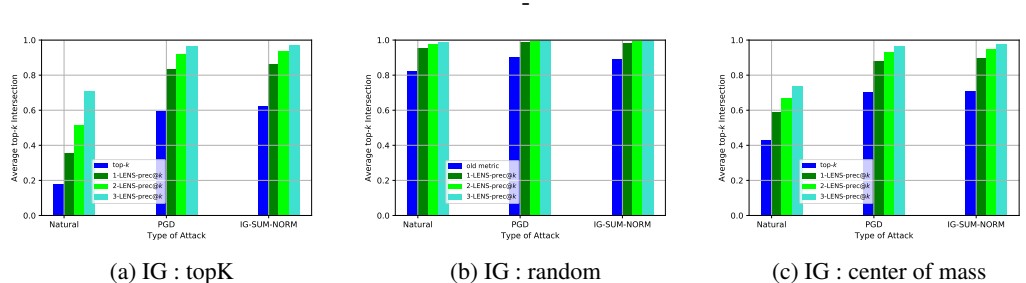

(a) IG : topK    (b) IG : random    (c) IG : center of mass

Figure 20: Attributional robustness of IG on naturally, PGD and IG-SUM-NORM trained models measured as top-$k$ intersection and $w$-LENS-prec@$k$ between the IG of the original images and the IG of their perturbations. Perturbations are obtained by the top-$t$ attack and the mass-center attack (Ghorbani et al., 2019) as well as a random perturbation. The plots show the effect of varying $w$ on Flower dataset.

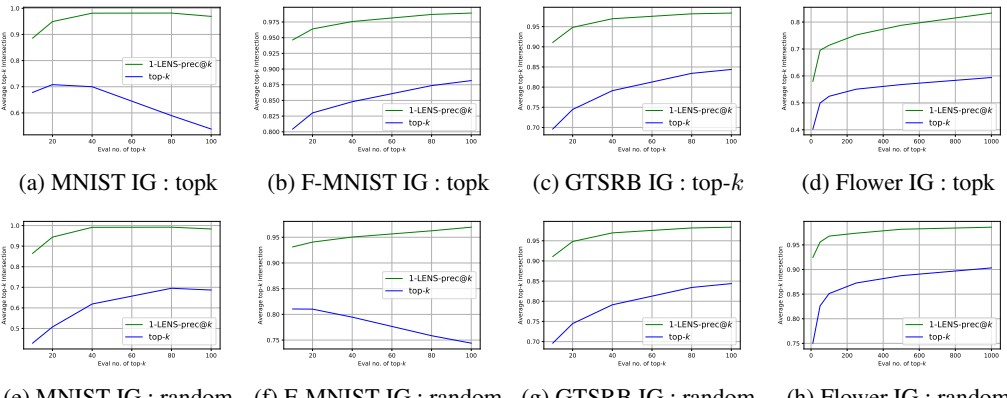

(a) MNIST IG : topk  (b) F-MNIST IG : topk  (c) GTSRB IG : top-$k$  (d) Flower IG : topk

(e) MNIST IG : random (f) F-MNIST IG : random (g) GTSRB IG : random (h) Flower IG : random

Figure 21: Comparing top-$k$ and 1-LENS-prec@$k$ with different $k$ for top-$k$ evaluation while the attack is a fixed $k$. For adversarially(PGD) trained network.

