# OpenReview forum: "Robust attributions require rethinking robustness metrics"
_ICLR.cc/2023/Conference — Submitted to ICLR 2023_

### Official Review · Reviewer_yyBB · 2022-10-21

**Confidence:** 4
**Correctness:** 2
**Technical Novelty And Significance:** 2
**Empirical Novelty And Significance:** 2
**Recommendation:** 3

**Clarity, Quality, Novelty And Reproducibility:**

The writing is clear in general and the proposed metrics for measuring attributional robustness are novel. However, the motivation for the new metrics is weak and the experimental evaluation is limited.

**Strength And Weaknesses:**

I think this paper has the following strengths:

1. The proposed metrics for measuring attribution robustness are novel. The idea of incorporating the locality of attributions along with their rank order when measuring attribution robustness is interesting.

2. It shows that the proposed LENS variants provide tighter bounds on the attribution robustness of already known improvements in attribution methods and model training designed for better attributions. This result seems significant.

3. The writing is clear in general and the related works are properly discussed.

However, I think this paper has the following weaknesses:

1. The motivations for the new metrics are not strong and need more explanations. One of the motivations is that under the existing metrics to quantify the attribution robustness of IG on a naturally trained model, even a single input-agnostic or universal random perturbation for all inputs can sometimes be a more effective attributional attack than using an independent random perturbation for each input. I think this motivation is weak since it is possible that the universal random perturbation is indeed a more effective attributional attack than the independent random perturbation. Also, this observation is not consistent across datasets (it only holds on the MNIST and Fashion MNIST datasets, but doesn't hold on the GTSRB and Flower datasets). I think for random perturbation attacks, it needs to repeat the experiments several times and report the mean and standard deviation of the results. It also needs to provide more justifications for the motivation.

2. In the proposed metrics, there is a hyper-parameter w. It doesn't discuss how to choose the hyper-parameter w. It seems the proposed metrics will be quite different for different values of w.

3. Based on the results in Figure 3, it claims that an attack may appear stronger under a weaker measure of attribution robustness if it ignores locality. I think this claim is not well supported by the results. It is possible that the top-t attack used is only effective for the top-k metric, but not effective for the proposed 1-LENS-recall@k and 1-LENS-prec@k metrics. For different metrics, it needs to design different adaptive attacks. Thus, the results in Figure 3 may be misleading.

4. Although it modifies the attack of Ghorbani et al. (2019) to construct a new attributional attack for the 1-LENS-prec@k objective, the experimental results show that the constructed attack is not effective. Since it doesn't provide the details of the constructed attack, it is hard for me to evaluate if the constructed attack is strong enough or not. It is possible that it doesn't design or implement the attack properly. Besides, I think constructing stronger attributional attacks for the proposed metrics should be included in this work, instead of future works. Without the adaptive attack evaluation for the proposed metrics, it is hard to know if the proposed metrics are reasonable or not.

**Summary Of The Paper:**

This paper shows the existing metrics for measuring attribution robustness like top-k intersection, Spearman's rank-order correlation, and Kendall's rank-order correlation are fragile. Specifically, it shows that under these metrics, a simple random perturbation attack can seem to be as significant as more principled attribution attacks. It then proposes Locality-sEnSitive (LENS) improvements of these metrics, namely, LENS-top-k, LENS-Spearman, and LENS-Kendall, that incorporate the locality of attributions along with their rank order. The proposed metrics provide tighter bounds on attribution robustness and do not disproportionately penalize attribution methods for reasonable local changes. The empirical results support the need for the new metrics proposed.

**Summary Of The Review:**

This paper proposes new metrics for evaluating attributional robustness. However, the motivation for the new metrics is weak and the experimental evaluation of the new metrics is limited. Thus, I recommend rejection.

---

### Official Review · Reviewer_UqJb · 2022-10-24

**Confidence:** 3
**Correctness:** 3
**Technical Novelty And Significance:** 2
**Empirical Novelty And Significance:** Not applicable
**Recommendation:** 3

**Clarity, Quality, Novelty And Reproducibility:**

**Clarity:** I think the clarity could be improved substantially. Currently, it is hard to parse how exactly the Locality-sENSitivity metrics are defined. For example, an overview figure could visualize the w-smoothed attribution, the LENS metrics, and the previous approach.

Furthermore, I found the notation quite a confusing. For example, T_k = S_k(x + \text{Att}(x)), but S_k and T_k are used without arguments in the definition of prec_k and recall_k on the same page.

**Quality:** The paper's presentation quality has to be improved: The figures' fonts need to be increased.

**Reproducibility:** Their approach is well described and the authors included a link to their anonymized source code.

**Novelty:** The paper has limited novelty: a smoothed version of existing metrics is proposed.

**Strength And Weaknesses:**

**Strengths:**

- This work tackles an important problem of how the robustness of attribution maps can be best measured.
- The connection between locality and robustness is an interesting observation.

**Weakness**:

- Recent work by Lundstrom et al. 2022 (ICML: https://proceedings.mlr.press/v162/lundstrom22a/lundstrom22a.pdf) found several theoretical limitations of IG. This questions the paper's assessment that *"Integrated Gradients (IG) is a standard attribution method based on solid theoretical foundations (Sundararajan et al., 2017)"*.
- More than only Integrated Gradients should be evaluated. Why is only this method is selected? Many other methods exist, also cited in the Related Work section. Different software packages provide these methods, so it should be straightforward to show more results. This might also create new insights as one method might score significantly different for the local-smoothed evaluation.
- The work is overall relatively incremental. Basically, it smooths existing metrics.
- The notation and clarity can be improved (see next section).


**Summary Of The Paper:**

Previous work has observed that imperceptible perturbations can manipulate the attribution maps. This work argues that previous metrics are in parts too sensitive to local changes. They propose smoothing the attribution map to weigh local changes less. Various metrics (top-k, Kendall's, and Spearman's rank-order) are then adapted to this setting. The experiments are all conducted on four different datasets using the Integrated Gradients method.

**Summary Of The Review:**

Overall, the paper is currently not ready for ICLR. The main reasons are the sole focus on Integrated Gradients and the lack of clarity, quality, and novelty.

---

### Official Review · Reviewer_mFur · 2022-10-28

**Confidence:** 4
**Correctness:** 2
**Technical Novelty And Significance:** 2
**Empirical Novelty And Significance:** 3
**Recommendation:** 3

**Clarity, Quality, Novelty And Reproducibility:**

Clarity: As mentioned in the previous section, the motivation behind the changes in the metrics is not very clear.

Quality: The evaluation of the paper could be significantly strengthened. Please see the detailed comments above.

Novelty: The main insight of considering the local neighborhood seems novel.

Reproducibility: The paper provides a link to the code though I did not inspect it.

**Strength And Weaknesses:**

Strengths:

 1. Assessing robustness of the metrics is an important topic.
 2. The insight that the top-k style evaluation metrics are insufficient is an important one.

Weaknesses:

Overall, the motivation behind the particular solution proposed in this paper is not very clear to me. The evaluation could also be improved further. Currently, the evidence seems to be that the results "make sense". Please see detailed comments and questions below.

1. First a clarification question. In Figure 2, how are the “random” and “universal random” different? Are they drawn from different distributions? Is universal random also scaled down to a l_inf norm of 0.3? I am a bit confused as to why random and universal random generate such a different effect if they are drawn from the same distribution.

2. The proposed solution of taking the local neighborhood of explanations into account does make some sense. However, is that the only problem with the robustness metrics? Are there any key issues that are still unresolved? My main concern here is that the insight made here seems quite isolated. The main motivation is presented in Figures 1 and 2 which has to do with considering the local neighborhood of the pixels. But it is not clear how much of the problem lies with the metrics and how much with the data domain (images, where adjacent pixels contain redundant information).

3. Building on the previous question, it is not clear how the insights would translate to other domains like tabular datasets or NLP.

4. Unfortunately, the paper does not compare different explainability methods so it is hard to disentangle how much of the problem is with the explainability method and how much with the metric. With a single explainer, it is not clear what a good reference value of the metric is.

5. Unlike the underlying task (classification), explainers do not have any ground truth. That makes the evaluation of robustness and evaluation of metrics quite difficult. Still, it is important to disentangle the weaknesses of metrics from weaknesses of explainers. I would suggest building toy tasks (e.g., https://arxiv.org/abs/2104.14403) where it would be easier to study these effects.

6. The design of the proposed metrics isn't quite discussed in sufficient details. How would considering the local neighborhood impact other related metrics like preservation game, pointing game (https://arxiv.org/abs/1910.08485)?



**Summary Of The Paper:**

The paper makes the points that current metrics to measure robustness of feature attributions not very effective and could raise false alarms. The paper then proposes a local neighborhood based smoothing procedure which essentially does not just consider the top-k feature attributions, but also the neighborhood around them. The paper then show how the improved metrics show higher overlap than their original versions. Further experiments are shown with attack-aware setups.

**Summary Of The Review:**

The motivation behind and the side-effects of the proposed changes in the explanation quality metrics are not clear. The evaluation focuses on one explanation method only so it is difficult to disentangle the effect of the explainer from the effect of the metric. Most of the evaluation results are of the kind "this makes sense". For these reasons, the paper does not seem ready for publication yet.

---

### Official Review · Reviewer_zTfh · 2022-10-29

**Confidence:** 2
**Correctness:** 3
**Technical Novelty And Significance:** 2
**Empirical Novelty And Significance:** 2
**Recommendation:** 5

**Clarity, Quality, Novelty And Reproducibility:**

The paper is written but the novelty and contribution of this work is small to the research community.

**Strength And Weaknesses:**

Strength:

The authors provide theoretically proofs to show that the proposed evaluation metrics provide tighter bounds on attributional robustness.

Weakness:

The experiment showing that random vectors are attributional attacks under existing evaluation metrics seems to be a bit confusing. If I understand correctly, it is not random vectors are strong attributional attacks but universal adversarial perturbations independent to each image. It would be good if the authors can clarify if these universal adversarial perturbations are optimized to be generated or randomly sampled.

In addition, the results in Figure 7 are less reasonable. According to "On Adaptive Attacks to Adversarial Example Defense" (NeurIPS-2020), the defense-aware attacks should be stronger than attacks without the knowledge of the defense. However, the results in Figure 7 display a reverse pattern of the conclusion in NeurIPS-2020, which makes me concerned that the designed defense-aware attacks are well-constructed.

Lastly, the datasets studied in this work are mainly small datasets, which makes it unclear how the conclusions/results be generalized to larger datasets, e.g., ImageNet, CIFAR, etc.





**Summary Of The Paper:**

This work shows that previous evaluation metrics for attributional robustness such as top-k intersections, spearman's rank-order correlation or Kendall's rank-order correlation are fragile where a simple random perturbation attack can seem to be as significant as more principled attributional attacks. Therefore, they propose locality-sensitive improvements of these metrics and show theoretically that the proposed metrics can provide tighter bounds on attributional robustness.

**Summary Of The Review:**

See above.

---

### Official Review · Reviewer_5YT2 · 2022-11-03

**Confidence:** 3
**Clarity, Quality, Novelty And Reproducibility:** The paper writing is easy to follow a…
**Correctness:** 3
**Technical Novelty And Significance:** 3
**Empirical Novelty And Significance:** 1
**Recommendation:** 3

**Strength And Weaknesses:**

Strengths:
- Authors conduct extensive literature survey and give sufficient context information of research.
- The experiments show the superiority compared to existing robustness metric. The consideration of locality sensitivity is reasonable.

Weaknesses:
- I am not convinced by the values of this work. AMs have been proven to have very minimal utility in previous works (e.g. [1,2]). The proxy metrics (e.g. robustness) are often ill-designed as they are not tied with the practical applications. [1] also pointed out that there is no correlation between AMs' effectiveness vs. their proxy scores in image classification.

[1] The effectiveness of feature attribution methods and its correlation with automatic evaluation scores, NeurIPS2021.

[2] HIVE: Evaluating the Human Interpretability of Visual Explanations, ECCV22.

Questions:
- Could authors justify why do we need the robustness for AMs. Attacks on defenses on AMs, in my opinion, are not realistic and trivial research problems.
- How this metric could help to assess the actual downstream utility of AMs?

I am happy to adjust the score once the authors satisfactorily resolve my concerns.

**Summary Of The Paper:**

This paper proposes a metric to measure the robustness of feature attribution methods (AMs) called LENS that better assesses the goodness of AMs. They are motivated by the insensitivity towards locality changes of existing robustness metrics for AMs.

While I found the metric is well-grounded and the experiments are sound, I believe this work has minimal impact in XAI then I lean towards Rejection.

**Summary Of The Review:**

My biggest concern lies on the practical impacts of this work and I think it does not fit ICLR.

---

### Decision · Program_Chairs · 2023-01-20

**Decision:**

Reject

**Justification For Why Not Higher Score:**

Please see concerns.

**Justification For Why Not Lower Score:**

NA

**Metareview: Summary, Strengths And Weaknesses:**

This paper proposes a metric to measure the robustness of feature attribution methods (AMs). They show that AMs such as top-k intersections, spearman's rank-order correlation or Kendall's rank-order correlation are fragile: under such metrics, a simple random perturbation attack can seem to be as significant as more principled attributional attacks. The authors propose Locality-sENSitive (LENS) improvements of the above metrics, namely, LENS-top-, LENS-Spearman and LENS-Kendall, that incorporate the locality of attributions along with their rank order. The authors theoretically show that the proposed metrics can provide tighter bounds on attributional robustness than their non-lens counterparts; experiments on benchmark datasets also support this.

Reviewers raised several concerns, e.g.:
- Earlier work indicates that proxy metrics may mislead the efforts in XAI (i.e. because they potentially cause ill-designed evaluations or unfair comparisons) and the proposed robustness metric is not an exception.
- There is still no clarity on which problems are caused by the metrics, and which are caused by the explainers and/or the data domain. Simply saying that "NLP is out of the scope" doesn't help since the same metrics and explainers are used in NLP too.  The paper should look into toy tasks (mentioned reviews) that generate "ground truth" for explanations to disentangle the effects.
- Recent work by Lundstrom et al. 2022 found several theoretical limitations of integrated gradients; the paper would be strengthened by considering other explainability methods as well.

**Summary Of Ac-Reviewer Meeting:**

NA